# The Impact of the COVID-19 Pandemic on Influenza Vaccination Attitudes and Actions in Spain’s Adult Population

**DOI:** 10.3390/vaccines11101514

**Published:** 2023-09-23

**Authors:** Camino Prada-García, Marina Toquero-Asensio, Virginia Fernández-Espinilla, Cristina Hernán-García, Iván Sanz-Muñoz, María Dolores Calvo-Nieves, Jose M. Eiros, Javier Castrodeza-Sanz

**Affiliations:** 1Department of Preventive Medicine and Public Health, University of Valladolid, 47005 Valladolid, Spain; vfernandeze@saludcastillayleon.es (V.F.-E.); chernang@saludcastillayleon.es (C.H.-G.); jjcastrodeza@saludcastillayleon.es (J.C.-S.); 2National Influenza Centre, Edificio Rondilla, Hospital Clínico Universitario de Valladolid, 47009 Valladolid, Spain; mtoqueroasensio@gmail.com (M.T.-A.); isanzm@saludcastillayleon.es (I.S.-M.); jmeiros@uva.es (J.M.E.); 3Dermatology Service, Complejo Asistencial Universitario de León, 24008 León, Spain; 4Preventive Medicine and Public Health Service, Hospital Clínico Universitario de Valladolid, 47003 Valladolid, Spain; 5Instituto de Estudios de Ciencias de la Salud de Castilla y León, ICSCYL, 24002 Soria, Spain; 6Centro de Investigación Biomédica en Red de Enfermedades Infecciosas (CIBERINFECC), 28029 Madrid, Spain; 7Department of Clinical Laboratory, Hospital Clínico Universitario de Valladolid, 47009 Valladolid, Spain; mdcalvon@saludcastillayleon.es; 8Microbiology Service, Hospital Universitario Río Hortega, 47012 Valladolid, Spain

**Keywords:** influenza vaccine, COVID-19, attitudes, Spain

## Abstract

Seasonal influenza is an acute respiratory infection caused by the influenza virus which constitutes a significant public health issue associated with high morbidity and mortality. The aim of this study was to investigate changes in attitudes, perceptions, and practices regarding influenza vaccination in the Spanish adult population during the COVID-19 pandemic, as well as their vaccination intentions, with special attention paid to those over 65 years old and in high-risk groups. To this end, a cross-sectional study was conducted through 2219 telephone interviews, and the results were compared with results obtained a year earlier. Regarding the reasons for deciding to get vaccinated in the 2022/23 season, a significant increase was observed in vaccine confidence (36.7% vs. 42.8%), social responsibility (32.5% vs. 43.8%), and in awareness of the importance of vaccination due to COVID-19 (21.7% vs. 25.4%). Advanced age (OR 2.8, 95% CI 2.0–3.9), belonging to high-risk groups (OR 2.7, 95% CI 2.0–3.7), and prior vaccination (OR 25.3, 95% CI 19.5–32.7) emerged as significant predictors for the intent to receive the influenza vaccine in the 2022/23 season. Continuously observing shifts in perceptions and behaviors related to influenza immunization is crucial to pinpoint factors that may influence the willingness to receive the vaccine and, in this way, design public health strategies that achieve a greater acceptance of it.

## 1. Introduction

Seasonal influenza is an acute respiratory infection, preventable through vaccination, which causes between 3 and 5 million severe illness cases worldwide and between 290,000 and 650,000 deaths annually [1,2]. This results in a significant strain on health services, making annual vaccination a key public health intervention to reduce morbidity, mortality, and the socioeconomic burden of seasonal influenza [3,4,5]. It has been shown that annual vaccination reduces the risk of death and complications in people aged 65 or older [6,7,8]. However, the acceptance of the influenza vaccine in most European countries remains low, despite the 75% target set by the World Health Organization and the European Centre for Disease Prevention and Control (ECDC) [9,10,11,12,13].

Since 2020, seasonal influenza seasons have coexisted with the COVID-19 pandemic, which has heightened the importance of the influenza vaccination, as co-infection with both viruses could pose an increased risk to the most vulnerable populations [14,15,16]. In this context, influenza vaccination coverage rates significantly increased in Spain, reaching values of over 65% in those over 65 years of age and 62% among healthcare workers [17].

COVID-19 and seasonal influenza both result in respiratory symptoms and have the potential to be fatal [18]. Elderly individuals and those with underlying health conditions, notably diabetes mellitus, hypertension, and chronic respiratory issues like asthma and chronic obstructive pulmonary disease (COPD), face heightened risks of severe complications and increased mortality [19]. Since the onset of the pandemic, the combined effects of influenza and COVID-19 have garnered significant attention. While dual infections are uncommon, they have been linked [20] to intensified respiratory distress. It is anticipated that upcoming vaccination campaigns may offer a single dose vaccine covering both viruses [21].

In September 2021, we conducted a representative cross-sectional observational study to understand the opinions, attitudes, and practices of Spanish citizens regarding influenza vaccination. We found that the COVID-19 pandemic may have had some repercussions on the national level of influenza vaccination coverage. In that same study, it was observed that the significant predictive factors of vaccination acceptance were age 65 or older, female gender, belonging to risk groups, and having been vaccinated previously at some point. Furthermore, the primary motivations for choosing vaccination were the desire for protection from the virus and the preference for yearly immunization. Conversely, the absence of a recommendation and the perception of the influenza vaccine as non-essential stood out as the leading causes for opting out of vaccination [10].

Similarly, in Italy, Domnich et al. [3] conducted a longitudinal study to assess changes in attitudes towards the influenza vaccination between the first and second pandemic waves of COVID-19, observing a significant increase in vaccine confidence.

Since the beginning of the pandemic in March 2020, new variants have emerged [22] and there have been changes in the epidemiology of influenza and SARS-CoV-2 infections, as well as in associated public health policies [23].

On 5 May 2023, the WHO COVID-19 Emergency Committee decided to declare the end of the emergency after considering the evolution and current data of the pandemic worldwide [24]. By the end of April 2023, over 765 million cases and more than 6.9 million deaths were reported globally. It is essential to remember that the COVID-19 pandemic still exists and poses a global threat, so epidemiological and virological surveillance must be maintained in light of its unknown future evolution [25].

During the 2022/23 season, circumstances were very different from the previous season, such as extensive COVID-19 vaccination, its declining incidence, and the removal of respiratory transmission containment measures. Therefore, the primary objective of this study is to monitor public opinion on influenza vaccination in relation to the evolution of the COVID-19 pandemic. In this document, we outline the shifts in attitudes, perceptions, and practices related to influenza vaccination among the Spanish adult population during the COVID-19 pandemic for the 2021/22 and 2022/23 seasons. We also delve into the factors promoting vaccination and the challenges hindering it, emphasizing the population aged 65 and above. By examining these elements, we aim to pinpoint key areas to further enhance the uptake of the influenza vaccine. This increased coverage would not only offer health benefits to those vaccinated but also provide broader societal benefits, including economic gains for the healthcare system.

## 2. Materials and Methods

### 2.1. Study Design

A national cross-sectional observational study was conducted during the 2022/23 season, for which an anonymous questionnaire in Appendix A, similar to the one used in 2021 [10], was employed to gather information on the perspectives, sentiments, and actions of Spanish adults concerning influenza vaccination in September 2022. This observational survey, rooted in participants’ feedback, adhered to the General Data Protection Regulation (GDPR) guidelines. Participation was on a voluntary basis, with GAD3, the overseeing firm, ensuring its proper execution and the privacy of the personal information of the respondents.

Subsequently, a comparative study was conducted concerning the results previously obtained during the 2021/22 season, conducting a repeated cross-sectional study to identify changes in the public opinion of Spanish citizens about influenza vaccination during the different stages of the COVID-19 pandemic.

### 2.2. Population under Study and Sampling Framework

The study covered all of the autonomous communities in its geographical scope and targeted individuals aged 18 and above. GAD3, a consultancy specializing in social research and communication [26], executed the study in collaboration with CSL Seqirus. Information was gathered using *Computer-Assisted Telephone Interviewing (CATI)*. Our interviewers were given a briefing to familiarize themselves with the objectives and vicissitudes of the study. This consisted of a group video call in which those responsible for the study (analysts) contextualized the study to be carried out, explained all the questions contained in the questionnaire, and gave technical instructions on how to formulate the questions and code the answers appropriately. The technical team was in contact with the interviewers throughout the fieldwork. No incentives of any kind were used. Contact was based on randomization and the voluntary participation of respondents in the study. Being a health-themed study, participation was higher than in other types of studies.

Non-probability sampling by quotas (sex/age) with simple sample allocation was carried out in all the autonomous communities of the national territory with the aim of providing regionalized results. The sample size consisted of 2219 random interviews, with an equitable distribution between landlines and mobile phones, conducted between 1st and 12th September 2022, from 9 in the morning to 9 at night. The questionnaire comprised 19 close-ended questions and each interview lasted approximately 4 to 5 min, similar to the methodology used in the previous year’s study [10]. The rejection rate of the total number of people contacted was 17.8%. The success rate (persons contacted who finally answered the survey in full) was 6.4%.

Participants eligible for the study were residents in Spain, aged over 18 years, who voluntarily agreed to participate. Those not meeting these criteria were excluded from the study. The sampling process adhered to quotas based on sex/age and geographic distribution, with a non-proportional allocation approach.

### 2.3. Questionnaire

The original questionnaire from 2021 [10] was slightly modified, adding some items was deemed important to study the changes in participants’ opinions during the COVID-19 pandemic concerning influenza vaccination. Demographic variables, such as age, gender, current employment status, education level, belonging to risk groups, and postal code of residence, were also recorded. The primary variable of interest was the intention to get vaccinated during the 2022/23 season. Both surveys also recorded vaccination attitudes in previous years, the most relevant reasons for deciding to get vaccinated or to not get vaccinated, the degree of information received about the vaccination campaign, the means through which they received any type of information, actions that can encourage vaccination, the importance of the influenza vaccine in the context of the COVID-19 pandemic, and the need for the administration to acquire a specific vaccine for the elderly.

### 2.4. Statistical Analysis

A descriptive statistical analysis was conducted using the data collected from the questionnaire responses. The study design had a margin of error of ±2.1% for a 95.5% confidence level (two sigmas) and was conducted under the most unfavorable hypothesis of P = Q = 0.5 in the assumption of simple random sampling. 

For the multivariate logistic regression analysis, the dependent variable was the response to the question “*And in view of the upcoming flu vaccination campaign, do you intend to get vaccinated?*”. The independent explanatory variables incorporated into the model were as follows:Belonging to a risk group.Dichotomous age (65 years or older, under 65).Dichotomous response (Yes or No) to the question: “*In the last few years, have you ever had a flu vaccination?*”

Variables were introduced based on their significance, and the *R*^2^ of the model was observed to select the best model. No goodness-of-fit test was conducted for the model. The association between variables was studied using Pearson’s chi-squared test. The odds ratios (ORs) of the studied variables were calculated using logistic regression with 95% confidence intervals. The significance level was set at *p* < 0.05. The analysis was carried out using the statistical software IBM SPSS Statistics version 25 (IBM Corp, Armonk, NY, USA) and the R Project for Statistical Computing for Windows.

## 3. Results

To present the findings of the study, the results are divided into three main sections. The first section provides details of the sociodemographic variables studied, the second refers to changes in the population’s attitudes and behaviors regarding influenza vaccination during the era of COVID-19, and the third section highlights the primary sources of information and strategies employed to encourage vaccination.

### 3.1. Sociodemographic Variables of the Study

Of the 2219 interviews conducted, 47.9% of the participants were men, and 52.1% were women. Regarding age, 25.9% were aged 65 or older. Regarding employment status, 29.5% were private sector workers, while nearly 28% were retired, and 13.6% were public sector workers. Additionally, in terms of education level, over 70% had completed secondary or university studies.

Regarding belonging to risk groups, 29% of respondents claimed to belong to a risk group. Of these, almost 50% were people over 65 years old, while 19.4% suffered from some form of respiratory pathology.

As previously observed, the distribution of the sociodemographic variables in this second survey correspond to the values obtained in 2021 [10] (Table 1).

### 3.2. Changes in Attitudes and Practices Regarding Influenza Vaccination

The analysis of previous vaccination rates showed that one-third of Spanish individuals got vaccinated every year, and up to 50% responded that they had never been vaccinated against influenza. Castilla-La Mancha, Murcia, and Madrid were the regions with the highest annual vaccination rates. Among people aged 65 or older, 72% got vaccinated annually, and up to 81.2% claimed to get vaccinated annually, frequently, or sporadically, compared to 38.4% of people under 65 years old (*p* < 0.001). Up to 80.9% of citizens belonging to a risk group got vaccinated annually, frequently, or sporadically, compared to 36.6% of those not belonging to any risk group (*p* < 0.001) (Table 2).

In our study, we examined the potential differences in responses between two seasons among the following groups: those belonging to a risk group, those not in a risk group, individuals younger than 65, those 65 or older, and the general population. Our findings indicate that, when compared to the previous season, there were no statistically significant differences in attitudes and practices related to prior influenza vaccination rates across the general population or any of the specific groups mentioned.

The main reasons Spanish individuals decided to get vaccinated against influenza were for personal and environmental protection (71.7%) and doctor recommendations (59.3%). A total of 19.7% responded that they got vaccinated against both influenza and COVID-19; this response was not available in the questionnaire conducted in the previous study. Based on age, the reasons “because my doctor recommended it” and “because it was recommended to me by nurses” were more frequently mentioned by people aged 65 or older than by the rest (*p* < 0.05), while those under 65 cited social responsibility more often than older individuals (*p* = 0.001). Regarding belonging or not belonging to risk groups, the reasons “for my own protection and/or my environment”, “because my doctor recommended it”, and “because it was recommended to me by nurses” were more frequently mentioned by those belonging to a risk group (*p* < 0.05).

Compared to the previous season, there was a significant increase in the information received about the influenza vaccine in the current season, a reason mentioned by 21.1% of respondents compared to 16.7% in the previous season. However, these differences were not found among people aged 65 or older (Table 3).

In total, 51.8% of Spanish individuals who had not been vaccinated against influenza in recent years stated that it had not been recommended/prescribed to them. This represents a 13-point increase compared to the previous season (*p* < 0.001). In the group of people aged 65 or older, the increase was almost 18 points, going from 30.3% to 48% (*p* < 0.05). Additionally, 32% of respondents believed that influenza was not severe, and the belief that there was a lack of information about the vaccine compared to the previous season increased from 10.2% to 14.9% (*p* < 0.001) (Table 3). This was not reflected among people aged 65 or older, which actually decreased from 12.2% to 10.4%, although not significantly (*p* = 0.926).

Of the Spanish population, 49% expressed an intention to get vaccinated during the 2022/23 influenza season. In the group of people aged 65 or older, this figure rose to 83.8%, a percentage similar to that found in the previous season, while only 39.3% of those under 65 years old expressed an intention to get the vaccine (*p* < 0.001). Residents of Madrid and Andalusia once again led in the highest intention to get the influenza vaccine (57% and 55%, respectively). Additionally, 83.4% of Spanish individuals belonging to a risk group expressed an intention to get vaccinated during the 2022/23 season (2 points higher than the previous year), compared to 37.4% of those not belonging to any risk group (*p* < 0.001). No significant differences were found between the previous season and the current one.

Of those who got vaccinated annually, 98.5% stated that they would get vaccinated again during the 2022/23 season. Among those who did so frequently or sporadically, the percentage intending to get vaccinated was 84.2% and 58%, respectively. However, only 15% of people who had never been vaccinated against influenza in the past expressed an intention to do so in the current campaign, 4% lower than the previous season (*p* < 0.001).

Regarding employment status, 80.1% of retirees intended to get vaccinated compared to 29.6% of students and 32.9% of private sector workers (*p* < 0.001). Compared to the previous season, there was an 8% decrease in vaccination intention among private sector workers and students, while there was a 12% increase among the self-employed/business owners.

As in the previous campaign, those with only primary education showed a higher intention to get vaccinated than those who had pursued higher education (*p* < 0.001) (Table 4).

The need to protect oneself from the virus (70%) and to get vaccinated annually against influenza (55.5%) were also the main reasons for vaccination during the 2022/23 campaign. Social responsibility increased by 11 points compared to the previous year (43.8%), being the third most important reason to get vaccinated (*p* < 0.001). Trust in vaccines accounted for 42.8% and grew by 6 points compared to the previous year (*p* = 0.004). The response “because COVID-19 has made me more aware of the importance of vaccination” also increased by 4 points (25.4%) compared to the last season (*p* < 0.05). Depending on age, the reason “it is necessary to get an annual flu vaccination” was more frequently mentioned by those aged 65 or older than by the rest (*p* < 0.001). In contrast, those under 65 years old cited a greater social responsibility and increased awareness of the importance of vaccination due to COVID-19 (*p* < 0.001). No significant differences were found in the responses from the age 65 or older group compared to the previous year. Regarding belonging to risk groups, the reasons “it’s necessary to protect against viruses”, “it is necessary to get an annual flu vaccination”, and “to be vaccinated against flu and COVID-19 at the same time” were more frequently given by those belonging to a risk group (*p* < 0.05).

However, the lack of prescription remains the primary reason for deciding not to get vaccinated (49.9%), increasing by 7 points compared to the previous year (*p* < 0.001). Additionally, up to 29.7% of those not getting vaccinated considered influenza not to be a severe virus, 6 points higher than the previous year’s response (*p* < 0.001), while 27.8% did not believe vaccination was necessary, also increasing from the previous year (*p* < 0.001). Regarding age, the lack of recommendation reached 41.4% within the group aged 65 or older, increasing by 10 points compared to the previous year (*p* = 0.12). In contrast, the reason “it’s not effective” was only given by 5.3% of older people, decreasing by 7 points compared to the previous year’s study (*p* = 0.046). Additionally, the reasons “I do not consider flu to be a serious virus” and “the COVID-19 vaccine is sufficient” were more frequently given by the elderly (45% and 14.6%, respectively) than by those under 65 (28.2% and 7.2%, respectively) (*p* < 0.05), while the lack of effectiveness was more frequently given by those under 65 (12.1%) compared to the group aged 65 or older (5.3%) (*p* = 0.046). Regarding belonging to risk groups, no statistically significant differences were observed in the given reasons (Table 5).

Regarding vaccination location preferences, 91% of Spanish individuals preferred to go to their health center. Additionally, 13.8% responded that they would like to access the influenza vaccine in their own home (in case of mobility problems), while 11% said that they preferred workplaces, hospitals, or specific vaccination areas. Based on age, people aged 65 or older preferred to go to their health center (*p* < 0.05), while younger individuals opted more for specific vaccination areas (such as those for the COVID-19 vaccination) and workplaces (*p* < 0.001).

Similar to the last season, as the respondents’ ages increased, the acceptance of a specific influenza vaccine for those over 65 decreased (*p* < 0.001). The percentage of those advocating for the public administration to purchase it, irrespective of cost, remained close to last year’s figures at 73.3%. However, acceptance among those aged 65 or older dropped from 72% to 64.4% compared to the last season (*p* = 0.05). Furthermore, nearly half (46.6%) of the Spanish population felt that the influenza vaccine gained significance amidst the COVID-19 pandemic, particularly among those below the age of 30. In contrast, the percentage of those 65 or older who felt that the influenza vaccine was of lesser importance reduced from 11% to 4.3% compared to the last season (*p* < 0.05).

Various variables were tested in our logistic regression models to identify predictors of the intention to get vaccinated during the 2022/23 campaign. The model that provided the best fit, with an *R*^2^ of 0.468, indicated that age, belonging to risk groups, and a history of prior vaccination were significant predictive variables. Accordingly, people aged 65 or older and those belonging to a risk group were 2.8 and 2.7 times more likely, respectively, to get vaccinated than those under 65 and those not belonging to any risk group. Furthermore, individuals who had been vaccinated in the past—whether annually, frequently, or sporadically—were 25.3 times more likely to get vaccinated during the 2022/23 campaign (Table 6).

### 3.3. Information and Measures to Promote Vaccination

Only 19.8% of the respondents claimed to have received information about the influenza vaccination campaign in the last year, although this figure doubles the percentage of those informed from the previous year (*p* < 0.001). Additionally, in the population group aged 65 or older, this figure rose from 7.4% to 17.8% (*p* < 0.001).

The main means of information about the vaccination campaign were healthcare professionals (39.5%), followed by traditional media (29.9%). Furthermore, 11.3% of the respondents claimed to have received the information at their workplace. However, there was a significant decrease in those who claimed to have received information from the public administration compared to the previous season (Figure 1). The primary source of information for people aged 65 or older was healthcare professionals (44.2%), with traditional information media decreasing from 53.3% last year to 43.3% in the current season. Additionally, it was observed that they still used social media at minimal percentages (1%) (*p* < 0.001). Furthermore, 49.6% of the people belonging to a risk group claimed to have received information about the vaccination campaign through healthcare professionals, compared to the 35% of the population that were not at risk.

The measures most appreciated to promote vaccination included providing information about the vaccine and its benefits (79.8%), ensuring easy access to primary care centers (76.6%), promoting the campaign through media outlets (70.7%), and sending annual reminder SMS (Short Message Service) messages (66.4%). Compared to the previous season, it is considered more important for people in the local environment to get vaccinated (*p* = 0.002), and to facilitate access to primary care centers (*p* = 0.022) and media awareness campaigns (*p* = 0.013). Whereas the creation of mass vaccination sites, like those for COVID-19, was considered less important this year (*p* < 0.001) (Table 7). In the group of people aged 65 or older, there was also a significant increase compared to the previous year in the valuation of the action “when people in my environment decide to get vaccinated” (*p* = 0.023) and a lower valuation of the measure related to the creation of mass vaccination sites like those for COVID-19 (*p* = 0.04). Furthermore, the measures related to people in the local environment getting vaccinated, media awareness campaigns, and facilitating access to primary care centers were preferred by people aged 65 or older compared to younger individuals (*p* < 0.05). Regarding belonging or not belonging to risk groups, all measures, except for the annual reminder SMS messages, were considered more important in people who belonged to a risk group (*p* < 0.05).

## 4. Discussion

Choosing whether or not to vaccinate involves a multifaceted decision-making process, shaped by factors like awareness, available information, societal norms, emotional responses, perceptions of risk, trust, and previous experiences. Furthermore, these aspects are likely to change over time and under different circumstances [28]. McDonald and the SAGE Working Group on Vaccine Hesitancy (WG) reviewed models about factors that can influence hesitancy and developed the Vaccine Hesitancy Determinants Matrix with factors grouped in the following three categories: contextual influences (historic, socio-cultural, environmental, health system/institutional, economic, or political factors), individual and group influences (personal perception of the vaccine or influences of the social/peer environment), and vaccine/vaccination-specific issues (directly related to the vaccine or vaccination). Finally, WG defined vaccine hesitancy as a “*delay in acceptance or refusal of vaccination despite availability of vaccination services and it is influenced by factors such as complacency, convenience and confidence”* [29]. Larson et al. conducted a systematic review of hesitancy around vaccines and concluded that the determinants of vaccine hesitancy are complex and context-specific, varying across time, place, and different vaccines [30].

This study was conducted to identify possible changes in the attitudes and perceptions of the Spanish adult population regarding influenza vaccination during the COVID-19 pandemic after one year. Additionally, vaccination location preferences were described, as well as variations in the most commonly used information media and measures to promote vaccination.

### 4.1. Perspectives and Behaviors Concerning Influenza Immunization

The previous vaccination rate showed figures slightly higher than those observed in the study conducted during the 2021/22 season, with no statistically significant differences found in the general population, or among people aged 65 or older or in risk groups [10]. Thus, about one-third of the respondents reported getting the influenza vaccine annually, with this figure being 72% for those aged 65 or older and reaching 81% for those belonging to a risk group.

Furthermore, 49% of the Spanish population expressed an intention to get vaccinated during the 2022/23 season, with 83.8% among those aged 65 or older and 83.4% in risk groups. These figures are similar to those of the 2021/22 season and show no significant differences [10]. However, only 15% of people who had never been vaccinated against influenza in the past expressed an intention to do so in the current campaign, 4% lower than the previous season (*p* < 0.001). This indicates a reduction in the positive reinforcement achieved during the early waves of the pandemic regarding the intention to get the influenza vaccine.

With the COVID-19 pandemic, the influenza vaccination coverage increased in all European countries [17,31], reaching 69.4% in Spain during the 2021/22 season and 64.6% during the 2022/23 season among people aged 65 or older [32]. This increase could help mitigate the negative effects of a higher prevalence of co-infections by both viruses [33,34]. However, greater efforts are still needed to achieve the vaccination coverage targets set by official organizations.

One of the possible reasons for the low coverage in the population is the false perception that influenza is a minor and low-risk disease [35,36,37,38,39]. Our results show that, in the non-vaccinated group, this factor increased from 23.2% to 29.7% when comparing data between both seasons (*p* < 0.001).

Trust in vaccines is another determining factor in vaccination coverage. Our study showed a significant increase in trust in vaccines among those who said that they intended to get vaccinated (*p* = 0.004), while distrust increased among those who did not intend to get vaccinated, rising from 10.9% to 14.8% (*p* < 0.001) when comparing the two seasons, a fact previously described by other authors [35,38]. In a recent study on the state of vaccine confidence in the European Union [40], a slight drop in confidence has been observed globally since 2020, however Spain maintains a high level of this parameter, surpassed only by Portugal.

The main reasons for wanting to get vaccinated during the 2022/23 season, as in the previous season, were the need to protect against viruses (70%) and the need to get the influenza vaccine annually (55%). Another reason present in the group intending to get vaccinated was social responsibility and protection of others, which emerged as a decisive factor for receiving the influenza vaccine and increased by 11 points when comparing the two seasons (*p* < 0.001), a fact observed in other qualitative studies [41,42].

The importance of prior vaccination as a determining factor in vaccination has been demonstrated by various studies [10,11,38,43,44]. In our work, this fact constituted a predictor factor for vaccination in the 2002/23 season (OR: 25.3; CI 95%: 19.5–32.7), so it is essential to promote this practice, especially among those not previously vaccinated.

However, the main reason for not getting vaccinated among those who did not intend to get vaccinated was the lack of a health professional recommendation for vaccination, which increased by seven points compared to the previous season (*p* < 0.001). Therefore, it is necessary to promote greater involvement and encouragement in recommending the vaccine to patients by healthcare professionals (doctors, nurses, pharmacists, etc.), as clinical evidence indicates that the recommendation of these professionals is the most effective measure to increase vaccination coverage in people with previous pathologies [35].

It is essential to identify the main reasons that lead the population to not want to get vaccinated, as refusing to get vaccinated is one of the top ten health threats, as reported by the WHO in 2019 [45], and it has been an even more challenging factor during the COVID-19 pandemic. Promoting greater involvement in recommending vaccination by healthcare professionals and improving attitudes and practices could increase vaccine acceptance against other diseases and represent a significant impact on public health [46,47].

### 4.2. Information and Measures to Promote Vaccination

The main source of information that respondents used to obtain information about the vaccination campaign was healthcare professionals, mainly among people aged 65 or older and those belonging to a risk group. This indicates trust in the knowledge and recommendations of healthcare professionals among respondents. Therefore, it would be important to implement information and training campaigns about influenza, its complications, and the effectiveness and safety of vaccination. There is a notable loss of trust in the public administration and social networks. The role of institutions is crucial in providing information about vaccination, but the observed loss of credibility indicates the need for a more appealing information format for the general population and content that is optimal in terms of comprehension. Regarding social networks, which saw their importance as an information medium reduced compared to the previous year, it is important to note that there is a lack of control over what is published, and they should be monitored by administrations, professional associations, and scientific societies to detect and combat the possible existence of “fake news”.

Similar to the study conducted last year [10], the most valued measure to encourage vaccination was to have more information, followed by better accessibility to primary care centers. The results suggest that efforts to boost influenza vaccine uptake should center around educational initiatives to enhance public understanding and appreciation of its significance. It is crucial for primary care physicians to utilize informative resources and convey straightforward messages highlighting the advantages of receiving the influenza vaccination [44]. This encouragement to vaccinate should also be present when any patient contacts healthcare personnel as part of their medical history. Additionally, it is necessary to improve accessibility to primary care centers, mainly among people aged 65 or older, as this is the preferred location for vaccination, especially in this age group, probably due to the close proximity to their residences.

Regarding the limitations of this study, it is important to highlight the possible memory biases derived from the responses collected directly from participants or the misinterpretation of questions by respondents. Additionally, the sampling may have been affected by a possible selection bias, meaning that those who did not respond might have different opinions about vaccination from those who did. Among the strengths is the fact that a representative sample of the population was obtained through a national survey. Weighted data were used to ensure better representation of the population.

## 5. Conclusions

Influenza during the COVID-19 pandemic and a history of previous vaccination are significant predictive factors for vaccination intent. The need to protect against the virus and to get vaccinated annually remained the main reasons for Spanish individuals to get vaccinated, while lack of prescription was the main cause of non-vaccination. All healthcare professionals and public health workers play a key role in communication and awareness about influenza and its vaccine, and any contact with the patient should be used as an opportunity to remind them of their vaccination schedule based on their age and underlying diseases. In addition, educational interventions should be implemented, and positive attitudes towards vaccination should be promoted, focusing on improving perceptions and correcting misinformation. There is also a need for a different institutional approach that allows for increased influenza vaccination coverage.

This study shows the changes that occurred during the COVID-19 pandemic and highlights the need to continue conducting opinion studies in the general population about the knowledge, attitudes, and practices surrounding the influenza vaccination, as well as information sources, to understand the trends and possible changes over time. This will allow us to develop effective interventions to improve vaccine acceptance both in the general population and in different risk groups. 

## Figures and Tables

**Figure 1 vaccines-11-01514-f001:**
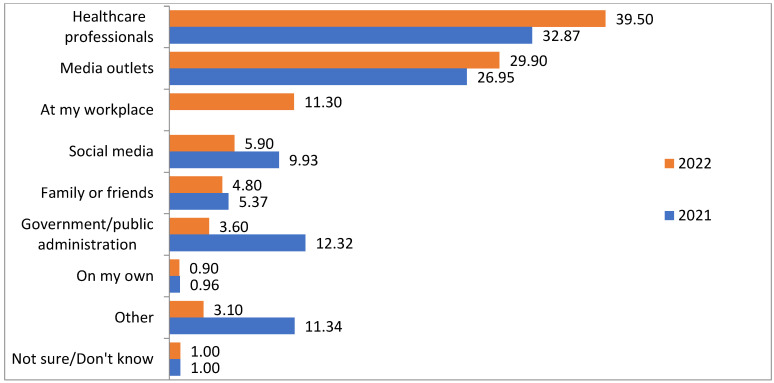
Primary sources of information on influenza vaccination for the adult demographic.

**Table 1 vaccines-11-01514-t001:** Sociodemographic characteristics of study participants.

Variables	2022 Survey N (%)	2021 Survey N (%)
**Gender**		
**Women**	1156 (52.1)	1139 (51.7)
**Men**	1063 (47.9)	1066 (48.3)
**Age Group**		
**18–29**	334 (15.0)	299 (13.6)
**30–44**	537 (24.2)	507 (23.0)
**45–64**	774 (34.9)	803 (36.4)
**65 or older**	575 (25.9)	596 (27.0)
**Employment Status**		
**Private sector worker**	656 (29.5)	659 (29.9)
**Public sector worker**	301 (13.6)	242 (11.0)
**Self-employed/business owner**	151 (6.8)	131 (5.9)
**Retired**	620 (27.9)	669 (30.4)
**Unemployed**	275 (12.4)	268 (12.1)
**Student**	72 (3.2)	94 (4.3)
**Homemaker**	146 (6.6)	142 (6.4)
**Education Level**		
**Primary or basic**	615 (27.7)	710 (32.2)
**Secondary ^1^**	932 (42.0)	869 (39.4)
**University**	672 (30.3)	626 (28.4)
**Belonging to Risk Groups** **People over 65 years old ** **Respiratory pathologies** **Cardiac pathologies** **Other chronic or immunosuppressed pathologies ** **Diabetics ** **Healthcare professionals ** **Immunosuppression ** **Essential workers ^2^ ** **Other ^3^**	321 (49.9) 125 (19.4) 80 (12.5) 71 (11.1) 57 (8.9) 43 (6.7) 33 (5.1) 34 (5.4) 96 (14.9)	249 (37.0) 115 (17.1) 79 (11.8) 66 (9.8) 19 (2.9) 87 (12.9) 27 (4.0) 17 (2.5) 14 (2.1)

^1^ Secondary education in Spain consists of compulsory secondary education and baccalaureate or vocational training degrees. ^2^ Essential workers are considered to be those indicated by the government Royal Decree-Law 10/2020 from 29th March published by the Government of Spain [27]. ^3^ Within the category “other”, there are three risk groups: people with disabilities, pregnant women, and people living with at-risk patients.

**Table 2 vaccines-11-01514-t002:** Attitudes and practices in relation to previous influenza vaccination level.

Variable	Values	Annual, Frequent, or Sporadic Vaccinations ^1^	
		N/Total (%)	*p*
**Risk group**	Yes	522/645 (80.9)	
	No	571/1558 (36.6)	<0.001
**Age**	≥65	467/575 (81.2)	
	<65	631/1642 (38.4)	<0.001

^1^ Annually (once a year), frequently (once every two years), sporadically (once every two years or more).

**Table 3 vaccines-11-01514-t003:** Reasons why people chose to get influenza vaccination or not in previous campaigns.

	Reasons	Survey 2021 N (%)	Survey 2022 N (%)	*p*
**Reasons for deciding to vaccinate**	For my own protection and/or environment Because my doctor recommended it Because of social responsibility Because I have suffered from the consequences of influenza in the past Because I have sufficient information about the vaccine Because it was recommended to me by nurses To get vaccinated simultaneously against influenza and SARS-CoV-2	755 (71.5) 642 (61.0) 296 (28.1) 180 (17.1) 176 (16.7) 130 (12.3)	770 (71.7) 633 (59.3) 319 (30.4) 181 (17.4) 220 (21.1) 128 (12.3) 204 (19.7)	0.922 0.399 0.240 0.859 0.012 1
**Reasons for deciding not to vaccinate**	Not recommended/prescribed My doctor has not recommended it I do not consider influenza to be serious I do not have enough information about the vaccine Lack of confidence in the effectiveness of the vaccine I have had a previous vaccination and it made me feel sick I have a phobia of needles	437 (38.6) 411 (36.3) 345 (30.5) 115 (10.2) 93 (8.2) 78 (6.9) 42 (3.7)	575 (51.8) 335 (32.0) 156 (14.9) 112 (10.7) 73 (6.9) 57 (5.4)	<0.001 0.516 <0.001 0.052 1 0.05

**Table 4 vaccines-11-01514-t004:** Attitudes and practices regarding intention to get vaccinated against influenza in the 2022/23 campaign during the COVID-19 pandemic.

Variable	Values	Intention to Vaccinate in the 2022/23 Campaign	
		N/Total (%)	*p*
**Risk group**	Yes	522/626 (83.4)	<0.001
	No	549/1467 (37.4)	
**Age**	≥65	469/560 (83.8)	<0.001
	<65	607/1545 (39.3)	
**Vaccination in** **recent years**	Yes, annually	711/722 (98.5)	
	Yes, with some frequency	48/57 (84.2)	<0.001
	Yes, sporadically	159/274 (58.0)	
	No	157/1051 (14.9)	
**Employment status**	Private sector worker	205/623 (32.9)	
	Public sector worker	137/278 (49.3)	
	Self-employed/entrepreneur	53/148 (35.8)	
	Retired or pensioner	483/603 (80.1)	<0.001
	Unemployed	104/260 (40.0)	
	Student	16/54 (29.6)	
	Unpaid domestic work	78/140 (55.7)	
**Level of studies**	Primary or lower	399/587 (68.0)	
	Secondary	389/880 (44.2)	<0.001
	University	288/638 (45.1)	

**Table 5 vaccines-11-01514-t005:** Reasons why people expressed that they were hesitant to receive influenza vaccination or not in the 2021/2022 and 2022/2023 campaigns.

	Reasons	Survey 2021 N (%)	Survey 2022 N (%)	*p*
**Reasons for deciding to vaccinate**	You need to protect yourself against viruses It is necessary to get an annual influenza vaccination I trust vaccines in general Because of social responsibility Because COVID-19 has made me more aware of the importance of vaccination. In case I have COVID-19, it will help me with the effects of the vaccine To be vaccinated against influenza and COVID-19 at the same time	748 (67.4) 596 (53.8) 403 (36.7) 358 (32.5) 239 (21.7) 212 (20.0)	734 (70.0) 567 (55.5) 435 (42.8) 444 (43.8) 257 (25.4) 270 (27.4)	0.206 0.428 0.004 <0.001 0.046
**Reasons for deciding not to vaccinate**	Not recommended/prescribed No vaccination is necessary I do not consider influenza to be a serious (deadly) virus I do not trust vaccines in general COVID-19 vaccine is sufficient Not effective	456 (42.6) 335 (31.7) 246 (23.2) 116 (10.9) 112 (10.7) 96 (9.1)	552 (49.9) 288 (27.8) 315 (29.7) 155 (14.8) 80 (7.7) 120 (11.4)	<0.001 0.05 <0.001 <0.001 0.018 0.064

**Table 6 vaccines-11-01514-t006:** Predictors of vaccination intention in the 2022/23 campaign.

Variables	Categories	p	OR	95% CI
**Age**	≥65 <65	<0.001	2.8 1	2.0–3.9
**Risk group membership**	Yes No	<0.001	2.7 1	2.0–3.7
**Previous vaccination**	Yes No	<0.001	25.3 1	19.5–32.7

**Table 7 vaccines-11-01514-t007:** Measures to promote vaccination.

Measures	Survey 2021 A lot + Quite N (%)	Survey 2022 A lot + Quite N (%)	*p*
**Having more information about vaccination and its benefits ** **Facilitating access to primary care centers** **A media awareness campaign ** **Sending annual reminders via SMS ** **People in my environment deciding to get vaccinated ** **Creation of mass vaccination sites like those for COVID-19**	897 (81.1%) 812 (73.9%) 746 (67.9%) 775 (70.1%) 586 (53.4%) 690 (62.3%)	1771 (79.8%) 1701 (76.6%) 1570 (70.7%) 1474 (66.4%) 1259 (56.7%) 1241 (55.9%)	0.67 0.022 0.013 0.1 0.002 <0.001

## Data Availability

The data presented in this study are available on request from the corresponding author. The data are not publicly available due to the privacy policies of the company that carried out the surveys.

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
