# Peer review of "The Impact of the COVID-19 Pandemic on Influenza Vaccination Attitudes and Actions in Spain’s Adult Population"

_vaccines, 2023, doi:10.3390/vaccines11101514_

Round 1

Reviewer 1 Report

Title:

The title suggests a study encompassing the entirety of the "Spanish adult population" which might give an impression of a nationwide scope. Given that the research specifically covers ten Autonomous Communities, it would be more accurate and transparent to reflect this in the title. A suggested revision could be: "... attitudes, perceptions, and practices regarding influenza vaccination in the adult population of ten Autonomous Communities in Spain during the COVID-19 pandemic."

Introduction:

The citations of References 1 and 2 in lines 450-451 do not directly support the claim made in line 40 about the disease's burden. Please cross-check and ensure a tight correlation between references and assertions.

The statement on the strain seasonal influenza places on health services needs further elucidation concerning References 3-5 to guarantee a clear link between the claims and supporting literature.

Line 43 presents a claim without a citation. Please attach a relevant reference for validation.

For the claim regarding influenza vaccine acceptance in European countries, it might be more appropriate to cite authoritative bodies like the WHO or ECDC. Also, the inclusion of Jordan, not being a European country, warrants reconsideration.

Reference 17's association with public health policies appears mismatched. Please ensure its relevance to the context.

Observe potential redundancy between lines 52-53 and 73-74. Consider merging these lines for brevity and clarity.

Methods:

In line 98, please elaborate on the validation status of the 2021 questionnaire. Additionally, provide more information on GAD3's data privacy measures.

The geographical scope described appears conflicting. Clarify whether the study covered all Autonomous Communities or specifically the ones listed.

On line 145, please provide:

The specific variables incorporated into the logistic regression model.

Any goodness of fit test conducted.

The criteria for variable inclusion.

Methodology Section Improvement Suggestions:

Scope Clarification: Clearly state that the study covered only ten Autonomous Communities, not the entire country. Also, explain the rationale for selecting these specific regions. Were they selected due to higher infection rates, diverse populations, logistical considerations, or other reasons?

Sampling Details:

Type of Sampling: Specify whether you used random sampling, stratified sampling, or another technique. This helps assess potential biases.

Sample Size: Indicate the sample size from each Autonomous Community and justify the chosen number. Was it based on population density, previous infection rates, or another criterion?

Questionnaire Validation:

Elaborate on the development and validation of the survey instrument. If it was adapted from a previous study, explain any modifications made.

Specify if the questionnaire underwent any reliability and validity testing (e.g., pilot testing, Cronbach's alpha calculation, etc.).

Include the questionnaire (or a supplementary link) for readers and other researchers to access.

Data Collection:

Detail the training given to the interviewers to ensure consistency and reduce biases.

Discuss any measures taken to increase the response rate (e.g., reminders, incentives).

If possible, provide the response rate, which can be vital to assess potential non-response bias.

Data Analysis:

Elaborate on the logistic regression model:

List the dependent and independent variables.

Specify any interaction terms or confounders you controlled for.

Detail any data transformations or standardizations.

Results:

For clarity, consider removing the introductory paragraph in lines 147-148.

In lines 161-162, to make the comparison meaningful, it might be helpful to include previous survey results in Table 1.

The phrasing in line 177 seems ambiguous. Please clarify whether you're indicating a 'reason' for vaccination or reporting an 'observation'.

Ensure your results' scope matches your study's actual reach. If data is primarily from ten autonomous regions, it might be appropriate to specify this.

The speculative nature of the statement in lines 214-215 might fit better in the Discussion section.

For clarity, consider rephrasing lines 302-303.

Discussion:

Reiterate caution when extrapolating results to the entire "Spanish population". The results should reflect the studied regions.

The suitability of reference 14, an opinion paper, should be reconsidered for supporting facts.

Address the ambiguous nature of lines 351-352 regarding a 'recent' 2007 study. It may be more beneficial to provide a current, relevant study or omit this line.

In line 366, please provide a more detailed description, suggesting "a health professional recommendation for vaccination".

One of my primary concerns is the absence of an encompassing theoretical framework to guide the research, such as the well-established Vaccine Hesitancy model. Incorporating such a framework could have facilitated a deeper, more structured understanding of the factors affecting vaccine uptake. It would have further allowed for a more comprehensive interpretation of findings and a robust comparison with similar studies in the literature.

It would add depth to the discussion to incorporate or reference established theoretical frameworks on vaccine hesitancy, like MacDonald's work or Larson's systematic review. This would ensure the research aligns with broader academic discourse on the topic.

MacDonald NE. Vaccine hesitancy: Definition, scope and determinants. Vaccine 2015;33:4161–4. https://doi.org/10.1016/j.vaccine.2015.04.036. Larson, H. J. et al. (2014). Understanding vaccine hesitancy around vaccines and vaccination from a global perspective: A systematic review of published literature, 2007-2012. In Vaccine (Vol. 32, Issue 19, pp. 2150–2159). Elsevier BV. https://doi.org/10.1016/j.vaccine.2014.01.081

  • The English language is comprehensible, but several passages would benefit from improved sentence structuring and word choice for clarity.

  •  
  • The manuscript does tend to be verbose in areas, making some sentences long-winded.

  •  
  • Here are the primary concerns:

    • Inconsistent Word Choices: There is some inconsistent terminology, such as the use of both "Spaniards" and "Spanish population".
    • Sentence Length: Several sentences are overly long and could be broken into shorter, clearer statements.
    • Wordiness: Some sentences can be condensed without losing their meaning.
    •  
    • Consistent Formatting: Number formatting, like "65 or older" vs "under 65 years old", should be consistent.
  • Recommendations for Improvement:

    • The manuscript would benefit from a thorough language edit to ensure clarity, consistency, and improved readability.
    • Consider using tables and graphs to present some of the data. This can break up the text and make the results more digestible for readers.
    • Ensure a consistent style for presenting statistics and percentage values.

In summary, while the content provided is valuable and provides insightful results, there is room for improving its presentation and clarity. Proper English language editing can make the manuscript more accessible and comprehensible to readers.

Reviewer 2 Report

This is a thorough, clear, informative and easily comprehensible report on attitudes and perceptions about flu vaccine in Spain, comparing two years related to the Covid pandemic.  You have done an excellent job reporting on your findings, changes over the two years, and you suggest appropriate interventions to improve acceptance of flu vaccine.  

There are few surprises in your findings.  The only surprise for this reader were the increased percentages of respondents providing reasons not to accept the flu vaccine. (table 5, bottom section).  This is of course concerning; your idea to increase training for medical care providers is an excellent suggestion.

Concerns:

1. Are you able to provide a response rate for the survey?  I assume it was low, but it would be helpful if you can say what percent of your calls were answered, what percent of persons who answered a call completed the survey, etc. 

2. Some minor word issues:

a. what are 'autonomous" communities.  Is that persons who are not institutionalized?  Please explain. (line 110)

b. In English we usually do not put "the" in front of the word "influenza". It is just "influenza" not "the influenza".  That is just convention. 

c. The first time you use the term SMS please spell it out. 

d. Line 304, "in the environment".  Do you mean, persons in the environment surrounding the respondent, or persons the respondent might come in contact with? This is unclear to me. 

e. Line 337: "vaccination coverage increased"... do you mean flu vaccination?

f. Line 411: I think "as" should be "is". 

Reviewer 3 Report

Introduction:

The manuscript delves into an intriguing topic, particularly pertinent due to the imperative to fortify vaccination strategies in light of the current epidemiological landscape. The study's primary objective was to investigate shifts in attitudes, perceptions, and behaviors concerning influenza vaccination among the Spanish adult populace amid the COVID-19 pandemic. It also explored vaccination intentions, with a specific focus on individuals aged 65 and older and high-risk groups. However, several aspects of the methodology and data analysis necessitate elucidation before considering publication.

Methods:

The authors mention conducting a comparative study by juxtaposing results obtained during the 2021/22 season with 2023, thereby converting an initial cross-sectional sample into a longitudinal study to discern evolving public sentiments among Spanish citizens. It is imperative to ascertain whether this study genuinely embodies the characteristics of a longitudinal study, particularly concerning the follow-up of participants.

While there is information about the sampling framework, there is a notable omission regarding the specific type of sampling employed. Additionally, crucial details about the variable used to determine the sample size are absent. It is essential to determine whether the study maintains representativeness at the autonomous community level to enable valid inter-community comparisons, as depicted in the study's results.

The documentation falls short of furnishing adequate information regarding the participation rate and the methodology employed by the GAD agency to select the sampled population. Furthermore, the inclusion and exclusion criteria require more comprehensive elucidation. It is also essential to discern whether any individuals declined to participate in the questionnaire and whether any incentives were offered to participants.

Questionnaire:

It is pertinent to inquire whether the authors conducted an assessment of the questionnaire's reliability, including whether Cronbach's alpha coefficient was calculated.

Statistical Analysis:

The authors have undertaken a multiple logistic regression analysis, but it remains unclear whether interaction effects were rigorously tested within the model.

Results:

The authors highlight that "over 70% had completed secondary or university studies" in terms of education level. It is essential to elucidate the precise meaning of "secondary studies" in the context of this study. Additionally, in Table 1, clarification is warranted regarding the term "essential workers."

The assertion that "no statistically significant differences were found in the general population or the previously described groups compared to the previous season" requires specification regarding the variables to which the authors are referring.

In Table 2, which presents comparisons of attitudes and practices relative to prior influenza vaccination levels in the 2022-23 survey, the total number of participants, as indicated in the study's results, amounts to 2219. However, some variable results do not align with this total.

The subtitle in Table 2, "Annual of frequent or sporadic vaccinations," lacks clarity.

While the results mention a decline in influenza vaccination rates among those aged 65 or older, dropping from 12.2% to 10.4% (Table 3), this age-based comparison is notably absent from Table 3. It may be beneficial to include a table comparing responses from individuals aged 65 and older with those from younger age groups, aligning with the study's objectives and results presentation.

The selection process for variables included in the logistic regression model remains unclear.

Tables and Figures:

A request for English translations of the subtitles in Table 7 is appropriate.

The absence of indications of statistically significant differences in Figure 1 between survey years should be addressed.

To enhance the analysis, consider adjusting comparisons between surveys by important variables, such as age and gender.

Discussion:

In the final paragraph of the discussion, the mention of data weighting to ensure better representation prompts the consideration of aligning data analysis with the study design. It appears that the statistical analysis was conducted under the assumption of a simple random sampling method in constructing the study group.

Conclusion:

Regarding the conclusion, it is advisable to assess whether the statement regarding extending health education initiatives to younger educational groups like schools and universities aligns with the study's findings and merits revision accordingly.

Moderate editing of the English language is required, Table 7 requires a translation in the heading. Some parts are repetitive please summarize
